# The Effect of Service Quality on Patient Citizenship Behaviors: Evidence from the Health Sector

**DOI:** 10.3390/healthcare11030370

**Published:** 2023-01-28

**Authors:** Saime Ulucayli, Kemal Cek, Adile Oniz

**Affiliations:** 1Faculty of Health Sciences, Department of Healthcare Organizations Management, Near East University, via Nicosia 99138, Turkey; 2Faculty of Economics and Administrative Sciences, Department of Accounting and Finance, Cyprus International University, via Nicosia 99258, Turkey

**Keywords:** service quality, patient satisfaction, patient loyalty, employee responsiveness, patient citizenship behavior

## Abstract

Background: Nowadays, health organizations seek to bring innovations to their services to stand out in competition with their rivals by improving service quality (SQ), encouraging patients to always make the same organizational choices, and enhance the behavior of patient citizenship. Objective: This study aims to determine the mediating role of patient satisfaction (PS), patient loyalty (PL), and employee responsiveness (ER) between the service quality and patient citizenship behaviors (PCB). Methods: In order to test the proposed hypotheses, quantitative research methods were utilized; cross-sectional data was collected using scales between December 2021 and March 2022. Results were obtained from 422 participants. The data were analyzed using descriptive statistics, correlation, confirmatory factor analysis, and structural equation modeling methods, using AMOS 21. Results: SQ was found to have a significant and positive effect on PL, PS, and ER. PL, PS, and ER were found to have a significant and positive effect on PCB. The indirect effect of SQ on PCB was found to be positive. Discussion: The findings demonstrate that SQ does not directly affect or create PCB, but it is affected by the mediators in order to create PCB via satisfaction, loyalty, and employee responsiveness.

## 1. Introduction

The age of technology and information that we live in allows many innovations to take place, including in public and private hospitals, which are an important part of the health sector. Thus, organizations try to create innovations in order to survive in a competitive environment. In this way, they seek to improve their services or products, both by researching how they can increase their market share while working to make the use of their services/products sustainable, and by satisfying the patient potential in the existing market. Organizations are not only affected by the emerging information and technological developments, but they also create some effects on the services and products offered. The most crucial need is to provide innovations which are related to health services. This includes service quality, satisfaction of patients with the service, and even how the service quality can be increased through the level of satisfaction or other constituents. In today’s circumstances, it is also emphasized that competition is present in the two main areas of service providers, which are public and private hospitals/institutions [1]. Nowadays, patients are more curious about sustainable and qualified treatment options within healthcare organizations. For that reason, both private and public hospitals focus on how superior they can be compared to others, by improving service quality and encouraging customers/patients to always make the same organizational choices, such as loyalty [2]. As a result, it is considered vital to manage the process of each innovation or development made by paying attention to these resources; since the main source of the development of enterprises such as health is technology, in addition to human resources, employees, and patients [3]. 

Patient citizenship behaviors (PCB) are considered necessary for both increasing the performance of businesses and benefitting other patients [4]. With PCB, it is clarified that the more patient and employee participation is ensured in businesses, the more they achieve values which avert threats by seizing existing opportunities [5]. Their strategies become distinctive according to the feedback from patients and enable them to obtain quality output in a competitive environment. For this reason, it is mentioned that PCB have a guiding quality in getting to know patients and discovering their expectations [4]. When PCB have been developed, a voluntary activity network is formed, and other patients are encouraged to adopt the same behaviors [6]. Hence, it is possible to reach high quality in the environment where the service is provided. It is emphasized that PCB have the same importance not only in concrete services, but also in sectors where intangible services are involved, such as the health sector, and that they are a guide which can reveal quality. It is highlighted that another important aspect of PCB is that they both cause loyalty and satisfaction towards an institution, and that they are also formed by these two outcomes. There is a mutual relationship between the two; as stated in the social exchange theory (SET), “the effect of causing and being caused” [7]. As stated in the literature, service quality (SQ), satisfaction, loyalty, and PCB are accepted as advantages for organizations in the competitive field. In fact, it has been identified that PCB have a positive relationship with perceived service quality and employee sensitivity in different types of fields [8,9]. Satisfaction, loyalty, and trust themes are the basis of achieving customer citizenship behaviors. At the same time, there are theoretical assumptions that PCB are formed as a result of the mentioned patient behaviors, such as satisfaction, loyalty, and trust [10,11]. Due to the rapidly increasing number of health institutions, the competition between these institutions intensifies. Patients are now receiving healthcare services from a variety of public/private healthcare institutions. Therefore, in such an environment, it becomes a necessity for institutions to act in accordance with the patient-oriented principle based on patient behaviors in order to establish and maintain a competitive advantage through achieving favorable patient behaviors. As a result of this, service quality and patient behavior issues have critical importance for institutions providing health services. From a patient-oriented perspective, the focus is that hospitals find ways to achieve favorable patient behaviors. Therefore, it is crucial to determine the factors contributing to the citizenship behavior of patients. Based on these advantages and mutual relations, in this study, we aimed to determine and compare the mediating role of patient satisfaction (PS), patient loyalty (PL), and employee sensitivity between service quality (of patients receiving health care services from both the public/private hospitals) and PCB. This study offers important contributions to the field. First, the dimension of PCB has the feature of being applied for the first time both in the field of health and in Cyprus. In addition, the concepts of SQ, PL, employee responsiveness (ER), and PS are applied together for the first time in the healthcare literature regarding these regions. Furthermore, the use of all the mentioned factors in a single study and the examination of the mediating relationship between them reveal the importance of this study.

In this study, we aimed to examine whether PL, PS, and ER mediate the relationship between SQ and PCB both in public and private hospitals. The results of the study showed that PL positively mediates the relationship between SQ and PCB. PS positively mediates the relationship between SQ and PCB. In addition, ER positively mediates the relationship between SQ and PCB. These findings aimed to shed light on achieving PCB within public/private hospitals. Achieving PCB is a source of competitive advantage for organizations, which can increase organizational efficiency by improving interactions between service participants. Furthermore, PCB are an important factor in maintaining a long-term customer–organization relationship through high-value service quality.

### 1.1. Theoretical Framework

It is evident from the following literature review that loyal patients/customers play an important role in the survival and development of organizations. In order to enhance “loyalty”, organizations have focused on SQ, which is another important reason for their existence, and that which has a relation to loyalty. SET is one of the effective theories for workplace behavior and it examines mutual relationships, such as “when one party benefits from another, the other party will feel responsible to return the favor at a future time” [12,13]. This theory emphasizes that interactions such as SQ and loyalty are interdependent to each other, and that they are always expected to create qualified relations which also lead to customer loyalty [14]. Furthermore, customer loyalty is the outcome of behavioral intention by a customer who is satisfied [15]. The SET relies on the relationship between benefits and costs [16]. When we focus on the relation between the healthcare organizations (public/private) and SET, it is emphasized that the patients mostly want to see social exchange as a behavior rather than economic exchanges [12,17].

Following SET, this study suggests that SQ affects PS, PL, ER, and PCB. There is a mutual relationship between the citizenship behaviors of patients and their perceptions on the service they receive from hospitals [18,19]. The quality of service is considered an essential tool for a hospital’s growth and existence. Service quality includes the patients’ observations, needs, and expectations, and whether those match with their actual behavior while they receive or use hospital services, when they visit a hospital within specific periods, and whether they are willing to do so due to the quality of services. This drives patient loyalty and satisfaction with the perceived quality of services [20,21]. Since good service quality drives patient satisfaction, patients will opt to visit the same hospital rather than others, which turns into behavioral actions, resulting in PCB. 

As a result, in this study, we integrated a model which explains how SQ directly affects PCB and also how it affects PCB through the mediators of PS, PL, and ER. First, the study investigates the mediating role of PS between SQ and PCB, then the mediating role of PL between SQ and PCB, and lastly, the mediating role of ER between SQ and PCB. The proposed relationship is presented in Figure 1 below.

#### 1.1.1. The Mediating Role of Patient Satisfaction

For the long-time survival of healthcare organizations in a competitive market, service providers focus on the term of “quality”. SQ measurements are used for supporting customers’ specific and unmet needs. For this reason, as most of the oldest definitions of quality include the “consumer’s judgement about a product or service about its superiority or excellence” [22], the basis of quality is related to the customer’s evaluation. Customer satisfaction and loyalty are also accepted as necessary to understand and develop quality. According to our study, PS is explained as the patient’s perception of whether a product or service meets their patient needs and expectations [23]. Based on that idea, customer satisfaction aims to ensure that “the overall performance of the service-product meets the expectations of customers” [24]. It can be said clearly that both quality and satisfaction focus on the needs and expectations of customers and that both have a high priority aspect for private/public healthcare service providers if they want to be successful in a competitive environment. Besides two strong variables that support each other, PCB also provide potential to the organizations in the competitive field. PCB define behavioral actions that customers engage in during/after service delivery with their own willingness [3,25].

Available studies explain that satisfied customers have more tendency to attend or engage in PCB. Studies posit the idea that engaging in PCB is a mutual behavior when the customers are satisfied. Additionally, satisfied behaviors are not the only way customers engage in PCB; customer loyalty, trust, and commitment are also actions involved in engaging in PCB [26]. Most studies show that PCB is affected positively by satisfaction, loyalty, and other mentioned mediators [3,27,28]. Satisfied customers or patients exhibit PCB such as giving recommendations, helping other patients/customers, or giving feedback to the organization [3]. Lengnick-Hall et al. clarified that there is a positive relationship between PCB and SQ [29]. Supportive results also come from Guo et al., supporting that SQ improves PCB, where customer satisfaction is improved by PCB [17]. Furthermore, the literature reveals that SQ affects satisfaction and creates PCB. Satisfaction improves individuals’ willingness or motivation to engage in PCB, where customers can offer positive and favorable results to the organizations [30]. Additionally, recent studies clarified that PCB are the combination of loyalty and satisfaction [17,31,32]. In this study, we developed the below hypothesis:

**H1.** *Patient satisfaction positively mediates the relationship between service quality and PCB (Figure 2 below)*.

#### 1.1.2. The Mediating Role of Patient Loyalty

Loyal customers are the individuals that choose or return to the same organization to buy services/products several times. Most hospitals prefer to use patient loyalty as a strategic plan in order to provide better service quality and have long-term customers/patients [33]. According to Oliver (1999), loyalty is to constantly repurchase a preferred product or service and to become a repeat customer of that product or service. PL, on the other hand, is the tendency to prefer the same hospital again if needed. It includes referring the services of the hospital positively to the people around and adopting the hospital as “my hospital”, since the patient likes the hospital services [34]. In order to be successful in patient loyalty, hospitals and service providers should meet the expectations and needs of patients, which is directly connected to service quality [35]. 

All hospitals try to set their principal goal as “successful and qualified growth”. According to the qualifications of growth and achievement, organizations need to first focus on quality, and then on satisfaction in services to build PL. For instance, when any organization, such as a hospital, improves their performance, profitability, and customer service quality, this directly has an effect on voluntary and extra role behaviors which are PCB [36,37]. This also provides benefits to the organizations where PCB have a relation with both existing and potential customers, as this is another dimension for growth [37]. In the study of Anbori et al., the relationship based on quality of service and loyalty has a significant effect on the patients’ willingness to return to the hospital [33]. Additionally, Hsiu-Yuan Hu et al. and Kuo et al. claimed that quality services influence customers’ satisfaction [38,39]. In addition to quality services that drive satisfaction, the perceived quality of service has direct and indirect effects on the behavior of customers. As permanent intentions/behaviors occur, hospitals need to ask themselves if the customer citizenship behavior is affected by the SQ and loyalty, and identify the role of PS on building PCB. Based on the mentioned hypothesis, this is supported by the studies of Revilla et al. and Mandl and Hogreve, which argue that there is a positive relationship between loyalty and PCB which results in “repurchase behavior” [31,32].

Hence, according to our research questions, we formed the hypothesis below:

**H2.** *Patient loyalty positively mediates the relationship between service quality and PCB (Figure 3 below)*.

#### 1.1.3. The Mediating Role of Employee Responsiveness

According to Setyawan et al., providing rapid and proper services to patients by responding to their complaints and their surroundings with appropriate information is the quality of “medical staff responsiveness”. In addition, the ability to provide appropriate and prompt services to patients by responding to their complaints or solving those complaints and supplying information to the patients is defined as the responsiveness of medical staff/employees [40]. SQ indicators highlight that in order to satisfy patients, hospital employees need to respond quickly and effectively to each patient’s complaints by offering satisfying explanations. Additionally, Cavena et al. and Kumar et al. claim that ER is related to customer satisfaction, which relies on understanding customers’/patients’ needs and wants [41,42]. ER is one of the moderators of satisfaction and loyalty [43]. There is a reciprocal relationship between the patients and the responsiveness of employees, which also leads and influences PCB. To have positive and long-term PCB, besides the dimensions of quality, satisfaction, and loyalty, a relationship between employees and customers is also necessary. Studies suggest that PCB rely on the social interaction between the customers and employees while they are receiving any service or perceiving the quality of services [27]. It is shown that patients evaluate the response of employees in the way it immediately concerns them. Tung et al. clarified in their study that to observe PCB, satisfaction and loyalty need to be used as moderators which determine ER [43].

**H3.** *Employee responsiveness positively mediates the relationship between service quality and PCB (Figure 4 below)*.

## 2. Materials and Methods

### 2.1. Sample and Data Collection

In order to test our proposed hypotheses, quantitative research methods were applied in this study. In line with the information received from the Northern Cyprus Statistical Institute, it was determined that the research population was 313,672 people. Since it would be difficult to reach the entire research population because of time, cost, and control limitations, the sample was selected using a stratified random sampling method to represent the research study population. Cross-sectional data were collected using scales between December (2021) and March (2022). Results were obtained from 422 participants who were 18 and older. Because of time limits and the restrictions of COVID-19 in the country (such as social distance rules and the quarantine process), it was not possible to reach the whole population. Based on these reasons, data collection was done face-to-face as well as via social media channels (Facebook, WhatsApp, etc.). Participants represent individuals from different regions of the country, and their eligibility to participate in the research was statistically evaluated. The data were collected from participants who represent different socio-demographic features and reside in the six different regions of Northern Cyprus. 

### 2.2. Instruments

In this study, we aimed to determine individuals’ satisfaction, loyalty, and employee responsiveness against the quality of services they received from hospitals, and to see if there is a mediating role played by the variables of satisfaction, loyalty, and employee responsiveness on PCB. In the study, a socio-demographic form which was developed and designed by the researchers was applied to the participants. Socio-demographic information about the respondents were collected, such as age, gender, education status, hospital preference (private/public), etc., which were included in the survey. The following framework continued, with the scales “Service Quality (SERVQUAL)” and “Patient Satisfaction” by Yesilyurt (Yesilyurt, O., unpublished data, 2018) [44], “Patient Loyalty” [45], “Employee Responsiveness” [46], and “Customer Citizenship Behavior” [6] applied to the participants. The “Service Quality Scale”—which was developed by SERVQUAL and includes 4 sub-dimensions with 22 items—is applied to the participants. Yesilyurt (Yesilyurt, O., unpublished data, 2018) developed the scale from Parasuraman et al. [47].

Additionally, the scale of Patient Satisfaction, also developed by Yeşilyurt (Yesilyurt, O., unpublished data, 2018), which occurred in 9 items with no sub-dimensions, was from [44,48] and Sututemiz (Sututemiz, N., unpublished data, 2005). Erdem et al. developed the scale of Patient Loyalty, benefiting from the patient satisfaction and loyalty studies of Oliver [34,45]. The scale consists of one dimension with 11 items. The Employee Responsiveness scale was developed by Nambisan et al. [46], consisting of 3 items. In order to measure patient citizenship behaviors (PCB), the Customer Citizenship Behavior (CCB) scale is used, which was translated by Aracı and Sezgin [49]; the original developers of CCB are Yi and Gong [6]. Aracı and Sezgin translated the scale into Turkish, and it consists of 4 sub-dimensions and 13 items. All scales used in this study are a 5-point Likert scale ranging from (1) Strongly Disagree to (5) Strongly Agree, and the scales have validity and reliability and they were approved by the developers (detailed Table 1 below).

### 2.3. Ethics

All the scales were approved by the “Near East University Scientific Research Ethics Committee”. In addition, informed consent forms were filled out by the participants both in electronic and written formats.

### 2.4. Demographics

The data were analyzed by using descriptive statistics, correlation, confirmatory factor analysis, and structural equation modeling methods using AMOS 21. The descriptive statistics of the collected data are shown in Table 2 below. Descriptive findings show that the participants of the study were 42.19% male and 57.8% female. The age groups of the participants were: between 18 and 25, 46.09%, between 25 and 35, 23.62%, between 35 and 45 17.80%, between 45 and 55 7.8%, and between 55 and 65, 4.14%. 52.68% of the participants had a university degree or higher and 38.04% of them graduated from high school. Participants in the study showed that 41.46% of them prefer to receive health services from public hospitals, 31.21% from private hospitals, and 27.31% prefer to receive health services both from the public and private sectors, mostly in combination.

The first dimension of SQ showed (n = 422) a mean score of 3.19 out of 5, which indicated that patients perceive a moderate level of SQ in public/private hospitals. The Physical properties—tangibles subdimension of SQ scores showed a mean value of 3.19. The Reliability subdimension of SQ scores showed a mean value of 3.23. The Responsiveness subdimension of SQ scores showed a mean value of 3.17 and the Assurance subdimension showed a mean value of 3.26. The last dimension of SQ, Empathy, showed a mean value of 3.11. 

The second dimension of the study, PS, showed (n = 422) a mean score of 3.25 out of 5, which indicated that patients perceive a moderate level of PL in public/private hospitals. The third dimension, PL, showed (n = 422) a mean score of 3.05 out of 5. ER showed (n = 422) a mean score of 3.19 out of 5. 

The last dimension, PCB, showed (n = 422) a mean score of 3.49 out of 5, which indicated that patients perceive a moderate level of PCB in public/private hospitals. The Helping other customers subdimension of PCB scores showed a mean value of 3.98. The Tolerating subdimension of PCB scores showed a mean value of 2.97. The Making recommendations subdimension of PCB scores showed a mean value of 3.25 and The Providing feedback to the organization subdimension of PCB scores showed a mean value of 3.61.

### 2.5. Reliability and Validity

To begin with, confirmatory factor analysis (CFA) was conducted to investigate the factor loadings and the validity of the constructs of the study. As a cut-off criterion, a 0.5 threshold was used for the construct loadings. Moreover, the model fit was tested using the following indices: the Comparative Fit Index (CFI), the Goodness of Fit Index (GFI), the chi-square mean/degree of freedom (CMIN/df), the root means square error (RMSEA), and the standardized root mean square residual (SRMR). The suggested threshold points for a good model fit should have the CFI and TLI above 0.90, RMSEA below 0.05, and SRMR below 0.09 (Hair et al., 2014). The model fit was found to be acceptable, as suggested by the fit indices (CMIN/df = 1.970, *p* < 0.05, Comparative Fit Index (CFI) = 0.95, Goodness of Fit index (GFI) = 0.90, root mean square error of approximation (RMSEA) = 0.04, and standardized root mean square residual (SRMR) = 0.05). 

To test the discriminant and convergent validity of the study, the average variance extracted and the squared correlation between the variables were checked and no issues were found. In addition, the single latent variable method was applied to test for the common method variance. The results implied that common method variance does not exist in this study. Composite reliability scores implied that the constructs are reliable and acceptable.

## 3. Results

### 3.1. Direct Effects

Table 3 below presents the direct effects between variables. SQ was found to have a significant and positive effect on PL, PS, and ER with β = 0.852, β = 0.882, and β = 0.821, respectively. Thus, considering H1, H2, and H3, PL, PS, and ER were found to have a significant and positive effect on the PCB: β = 0.586, β = 0.424, and β = 0.177, respectively. In addition, SQ was found to have a negative effect on PCB (β = −0.180).

### 3.2. Mediating Effects

The mediation and indirect effects of PL, PS, and ER (Table 4 below) were investigated by employing 95% bias-corrected bootstrapped confidence intervals (N = 5000). The results of the mediation analysis showed that PL positively mediates the relationship between SQ and PCB (β = 0.253, *p* < 0.05). Thus, H1 is accepted. PS positively mediates the relationship between SQ and PCB (β = 0.143, *p* < 0.05). Thus, H2 is accepted. Lastly, ER positively mediates the relationship between SQ and PCB (β = 0.048, *p* < 0.05). Thus, H3 is accepted.

## 4. Discussion

This study investigated the relationship between SQ and PCB through the mediating dimensions PS, PL, and ER, based on the SET within private/public hospitals. The findings demonstrated that SQ does not directly affect or create PCB, but it is affected by the mediators to create PCB via satisfaction, loyalty, and employee responsiveness. The findings validate the SET, which states that PCB are behavioral outcomes of the relational exchange in health services. This outcome is interrelated with loyalty, satisfaction, and commitment, which are supportive for building PCB, which is also supported by Robertson et al. [50]. Furthermore, Patterson and Razzaque [51] agree that the SET is a basis for the link between satisfaction and PCB. There are some actions that patients display voluntarily, and these actions actually emerge and become widespread with the benefits they receive from the other side (organization). However, the SET does not explain the economical exchanges from health services, which are short-term; they rely on social exchanges such as loyalty, satisfaction, and, lastly, PCB [52,53]. It is suggested by previous studies that the social exchange model is effective in relationships, considering satisfaction, loyalty and trust. It is also suggested that they can act as mediating variables for PCB [18,54].

On the other side, the health sector, including public/private hospitals and healthcare, consists of complex structures and services (regarding the diversity of tasks and duties that are delivered to the patients). With the continuous improvement in health systems and reforms, the health services market is slowly opening up and medical institutions are facing more and more fierce market competition. This competition makes both public and private organizations maintain sustainable development only by ensuring satisfactory medical services to patients, winning the patients’ trust in competition with many medical service institutions, and, finally, by obtaining large numbers of loyal patients.

The findings of this study posit that SQ is not directly influenced by PCB positively; it shows that PS, PL, and ER have a mediating influence between SQ and PCB, as also mentioned above by the SET. It is considered that quality leads firstly from satisfaction and loyalty. Therefore, loyalty and satisfaction are affected by qualified services. Patients have expectations before they visit a health institution. If these expectations are satisfied, they will regularly visit the organization out of loyalty, and they create PCB [23]. The literature also supports that service quality influences PCB through patient satisfaction. Quality is formed on the basis of creating satisfaction with firms/health institutions, which leads to patient–firm-based relations. This creates a relationship between the firm quality and satisfaction, which demonstrates PCB to the firms/health institutions [26,55]. PS is the basis of PL, and ER has a direct relationship with SQ. Quality service leads to satisfied patients. Good care, communication, and the attitude of employees in helping the patients all influence their satisfaction with the services. There is a multidimensional construct of service quality where the patients judge their interactions with the service providers and services [55,56].

From these judgements, strong connections develop between the quality of service and PS [57]. Due to this connection, satisfied patients have the tendency to return to a hospital, and the results show a significant link between SQ and PS and loyalty [58]. Past studies have also shown that quality of service directly affects loyalty, and that satisfaction also results from the quality of service [59,60,61,62]. In our findings, it was shown that PS positively affects PL. This finding is also supported by the studies in the literature [63,64,65]. However, in the studies of Liu et al. and Hu et al., no statistical significance was found between PS and PL [1,38]. The literature claims that a loyal patient is one who is satisfied and always aims to obtain health services from the same organization; therefore, the patient develops that relation through their satisfaction. A satisfied patient is a loyal patient [63,66].

ER is also found in our study to have a mediating effect on PCB through SQ. Patients mostly have the expectation to be well cared for, both before and after they visit the health institution. Most of the participants claimed that “the employee(s) of the hospital they receive services from takes immediate action to address their concerns” and “the employee(s) of the hospital where patients receive services are sensitive to their suggestions and requests”. These findings are also supported by other studies citing that employee responsiveness has a relationship with the patient’s expectations of their health care needs [58,67].

The pandemic circumstances and current restrictions in the country have created limitations in reaching the entire population. For these reasons, participants were included in the research using the method of stratified random sampling. Participants represented individuals from the different regions of the island and were evaluated statistically to be able to participate in the research. Additionally, due to the pandemic-related restrictions, the data collection was mostly performed through online questionnaire platforms.

Since the customer citizenship behavior in this study was used for the first time in the field of health, there were limitations in similar studies in the literature. This study is the first to evaluate the relationship between the patient citizenship behaviors and service quality in the health sector. In adding a new dimension to the healthcare field, being new and first in the literature was a limitation. While all these factors posed obstacles for the researcher to access more data, reconsidering face-to-face interviews in future studies may enable us to reveal different demands, since participants’ reactions and expectations towards PCB were not questioned. In addition, the roles of different mediating variables, such as the service quality, patient loyalty, patient satisfaction, and PCB, each carry a potential for further research. This study can be used by: (1) the government as an evaluation of public services and the private healthcare sector, (2) hospitals, as information material to make decisions and improvements, and (3) academics, as reference for further research. Researchers can add further indicators, and can also develop this research by using other tools of analysis and different research objects. In addition, the roles of different mediating variables, such as the service quality, patient loyalty, patient satisfaction, and PCB, each carry potential for further research.

## 5. Conclusions

The literature supported our claim that PCB depend on service users’ perception of quality and employees’ reactions during service encounters [27]. It is also stated that service users expect immediate actions from employees if they have any problems or suggestions when checking their immediate responsiveness [43]. As it is significantly seen in our study, Yi et al. and Yi and Gong [26,36] support that PCB also have a positive relation with ER. If patients have higher PCB, employees also build their loyalty to the organization, as explained by the cited studies.

This study contributes to the relation between SQ and PCB in the literature by providing insights into the mediator effects of PS, PL, and ER. Although there is a growing number of studies conducted on the relation of SQ and PS, PL, and ER, the PCB and mediator relations still require deeper research. Another unique contribution of this study is that it is the first known study to investigate the mediating effects of service quality on patient citizenship behaviors in the health sector. The findings show that service quality affects patient satisfaction, patient loyalty, and employee responsiveness, which in turn contribute to and affect PCB.

This research has several implications for regulatory organizations and healthcare management. This study demonstrates a strong relationship between employee responsiveness, patient loyalty, and patient satisfaction and the citizenship behavior of patients. Therefore, it is suggested that healthcare managers focus on activities that promote the healthcare employees to be more responsive towards their patients and to participate more proactively in activities that promote the satisfaction and loyalty of patients. In the absence of these, service quality alone may have a negative or no influence on citizenship behaviors. Thus, it is crucial for healthcare managers to achieve satisfaction, loyalty, and higher employee responsiveness. It can be suggested that managers take into account the elements that contribute to these factors within the healthcare industry. Managers should take into account the effect of service quality on the patient when creating specialized quality and marketing plans that can aid in developing long-lasting loyalty–satisfaction–citizenship ties with patients. These methods may include a range of topics, such as information management and communication, service packages, payment styles, price policy, and other cutting-edge quality and marketing guidelines. However, healthcare facilities must not ignore their moral and legal obligations while trying to achieve these behavioral factors. Furthermore, regulatory bodies should be in charge of supporting individuals in evaluating both their present and alternative healthcare providers efficiently.

## Figures and Tables

**Figure 1 healthcare-11-00370-f001:**
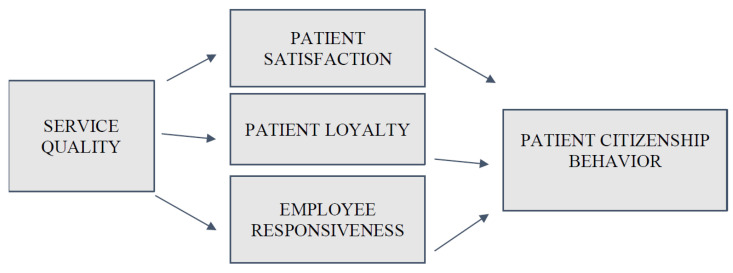
Research Model.

**Figure 2 healthcare-11-00370-f002:**
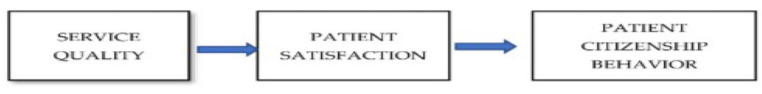
Mediating Effect of Patient Satisfaction.

**Figure 3 healthcare-11-00370-f003:**
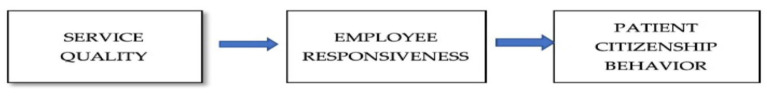
Mediating Effect of Patient Loyalty.

**Figure 4 healthcare-11-00370-f004:**
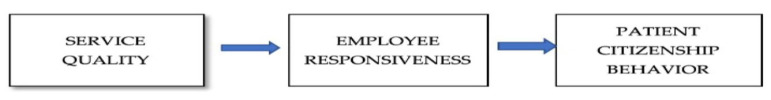
Mediating Effect of Employee Responsiveness.

**Table 1 healthcare-11-00370-t001:** Scale Descriptions.

Scale	Sub-Dimension	Questions	Authors
**Service Quality**	Physical properties—tangibles	1–4	(Yesilyurt and Tekin, 2018) [44]
	Reliability	5–9	
	Responsiveness	10–13	
	Assurance	14–17	
	Empathy	18–22	
**Patient Satisfaction**	-	9 questions	(Yesilyurt and Tekin, 2018) [44]
**Patient Loyalty**	-	11 questions	(Erdem et al. 2008) [45]
**Employee Responsiveness**		2 questions	(Nambisan et al., 2016) [46]
**Patient Citizenship Behaviors**	Helping customers	1–4	(Aracı and Sezgin, 2020) [49]
	Flexibility—tolerance	5–7	
	Making recommendations	8–10	
	Providing feedback to the organization	11–13	

**Table 2 healthcare-11-00370-t002:** Descriptive Analysis of Participants.

		Number of Participants (N)	%
Sex	Male	173	42.19
	Female	237	57.80
	**Total**	**410**	**100**
Education	Primary School	32	7.8
	Secondary School	6	1.4
	High school	156	38.04
	University or higher	216	52.68
	**Total**	**410**	**100**
Age	18–25	189	46.09
	26–35	99	23.62
	36–45	73	17.80
	46–55	32	7.8
	56–65	17	4.14
	**Total**	**410**	**100**
Region	Nicosia	176	42.92
	Kyrenia	39	9.5
	Famagusta	124	30.24
	Iskele	34	8.29
	Lefke	13	3.1
	Guzelyurt	24	5.85
	**Total**	**410**	**100**
Participant’s Hospital Preferences	Public Hospital	170	41.46
	Private Hospital	128	31.21
	Both Public and Private Together	112	27.31
	**Total**	**410**	**100**

**Table 3 healthcare-11-00370-t003:** Direct Effects Between Variables.

Parameter			Estimate	Lower	Upper	P	Hypothesis	Accepted/Rejected
SQ	→	PL	0.852	0.808	0.897	0.010	H2	Accepted
SQ	→	PS	0.882	0.844	0.916	0.010	H1	Accepted
SQ	→	ER	0.821	0.768	0.878	0.010	H3	Accepted
PL	→	PCB	0.586	0.418	0.717	0.010	H2	Accepted
PS	→	PCB	0.424	0.264	0.595	0.010	H1	Accepted
ER	→	PCB	0.177	0.088	0.303	0.010	H3	Accepted
SQ	→	PCB	−0.180	−0.347	−0.063	0.018		

Notes: SQ, Service Quality; PL, Patient Loyalty; PS, Patient Satisfaction; PCB, Patient Citizenship Behaviors; ER, Employee Responsiveness; P, significance.

**Table 4 healthcare-11-00370-t004:** Mediating Effects Between Variables.

Parameter	Estimate	Lower	Upper	P	Hypothesis	Accepted/Rejected
SQ --> PS --> PCB	0.143	0.088	0.207	0.010	H1	Accepted
SQ --> PL --> PCB	0.253	0.168	0.350	0.010	H2	Accepted
SQ --> ER --> PCB	0.048	0.011	0.097	0.025	H3	Accepted

Notes: SQ, Service Quality; PL, Patient Loyalty; PS, Patient Satisfaction; PCB, Patient Citizenship Behaviors; ER, Employee Responsiveness; P, significance.

## Data Availability

The data presented in this study are available on request from the corresponding author, upon reasonable request (saime.ulucayli@neu.edu.tr).

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
