# Peer review of "The Effect of Service Quality on Patient Citizenship Behaviors: Evidence from the Health Sector"

_healthcare, 2023, doi:10.3390/healthcare11030370_

Round 1

Reviewer 1 Report

Using satisfaction, loyalty, and employee responsiveness as mediators, this paper shows that service quality does not directly affect or create PCB. 

1) The suggested author in the abstract follows the background, objective, methods, results, and discussion. 

2) Suggested the authors rewrite the introduction and find more significant to the 

1I the correlation of the effect of service quality on patient citizenship behaviors: 2) Evidence from the Health Sector. 

Which sector is your target in this manuscript, and reason? 

3)Lines 89-92," According to the discussed variables and the results...." What are the variables and the results?

4) What is within the health sector in line 92? 

5) In line 269,  Descriptive Analysis of Participants. Age 25,35,45 and 55 is deprecated count. 

6) Explain the results of SQ → PCB, -0.063, the difference between the results and the Estimate and the Upper. What are the results behind it?

7) Specifically explain the "On the other side, the health sector consists of complex structures and services. " The health sector is broad, suggested to choose the health sector more specific. 

8)"y medical service institutions, and 332 finally by obtaining large numbers of loyal patients. This study examines the research 333 questions if the SQ influences PCB by the mediating effects of patient satisfaction, patient 334 loyalty, and employee responsiveness within the healthcare institution" I suggested moving this wording to the abstract/ introduction. 

9)This is a request to attach the data and for the reviewer to double-check the results. 

10)Suggested adding the limitation and future implications. 

Author Response

Reviewer 1:

Reviewer’s Comment:

The suggested author in the abstract follows the background, objective, methods. results, and discussion.

Authors’ Response:

Abstract has been updated according to the suggestions.

Reviewer’s Comment:

Suggested the authors rewrite the introduction and find more significant to the

1) the correlation of the effect of service quality on patient citizenship behaviors: 2) Evidence from the Health Sector

Which sector is your target in this manuscript, and reason?

Authors’ Response:

Introduction has been improved and health sector is specified and explained.

Reviewer’s Comment:

Lines 89-92," According to the discussed variables and the results... What are the variables and the results?
Authors’ Response:
variables and results were added.

Reviewer’s Comment:
What is within the health sector in line 92?
Authors’ Response:
health sector is specified.

Reviewer’s Comment:

In line 269, Descriptive Analysis of Participants. Age 25 35,45 and 55 is deprecated count.
Authors’ Response:
The table has been fixed.

Reviewer’s Comment:

Explain the results of SQ   PCB, -0.063, the difference between the results and the Estimate and the Upper. What are the results behind it?

Authors’ Response:

The findings have been explained. In the absence of the mediator the service quality may have a negative influence. However, the mediators help the effect to be positive between service quality and the dependent variables.

Reviewer’s Comment:

Specifically explain the "On the other side, the health sector consists of complex structures and services." The health sector is broad, suggested to choose the health sector more specific.
Authors’ Response:
health sector have been specified.

Reviewer’s Comment:

y medical service institutions, and 332 finally by obtaining large numbers of loyal patients. This study examines the research 333 questions if the SQ influences PCB by the mediating effects of patient satisfaction, patient 334 loyalty, and employee responsiveness within the healthcare institution" I suggested moving this wording to the abstract/Introduction.

Authors’ Response:
moved to the introduction.
Reviewer’s Comment:
This is a request to attach the data and for the reviewer to double-check the results.
Authors’ Response:
Data files cannot be uploaded. 

Reviewer’s Comment:

Suggested adding the limitation and future implications.
Authors’ Response:
limitation and future implication have been added.

Reviewer 2 Report

Thank you for the opportunity to review the manuscript ‘The Effect of Service Quality on Patient Citizenship Behaviors: Evidence from the Health Sector’.

The authors conducted the study to determine and compare the mediating role of patient satisfaction, patient loyalty, and  employee responsiveness between the service quality and patient citizenship behaviors. This manuscript presents useful and interesting findings which help to understand the relationship between service quality and patient citizenship behaviors, with both practical and theoretical implications considering that patients are widely acknowledged as indispensable co-creators of value in the healthcare context.

However minor revisions are needed. Some particular issues which need to be addressed are shown below.

Introduction section

·        Please add abbreviations for the following terms: social exchange theory, patient satisfaction, patient loyalty, employee responsiveness, when first mentioned in the manuscript,

·        In order to make it easier for the reader to follow the research hypotheses, it is worth considering to present in subsections 1.1.1, 1.1.2, 1.1.3 in figures the relationship/interrelationship between the particular components of the research model

·        It would be useful to highlight the relevant definition of terms in each of the subsections given above: i.e. patient satisfaction, patient loyalty, employee responsiveness

Materials & Methods section:

·        Information should be added on the form in which the survey was conducted: online survey or questionnaire survey, and what was the sampling method

·        Please consider adding a table/statement with the research tools used, describing:

o   measured attribute: Service Quality; Patient Satisfaction; Patient Loyalty; Employee Responsiveness; Customer Citizenship Behavior

o   author of the tool, year of last version of the tool

o   names of dimensions/sub-dimensions, number of items of particular scale

Results section:

·        Age groups are not precisely defined, i.e. overlapping (e.g. respondents aged 25, 35, 45, 55)

·        It is worth supplementing the results section with results from individual scales concerning, for example, patient loyalty or patient satisfaction

Discussion section:

·        Limitation of the study should be moved from the Conclusions section to the Discussion section.

Conclusions section:

·        It is important to emphasise the practical application of the results obtained.

References:

·        The reference list needs formatting in accordance with the guidelines.

Author Response

Reviewer 2:

Introduction section:
Reviewer’s Comment:
Please add abbreviations for the following terms: social exchange theory, patient satisfaction, patient loyalty and employee responsiveness when first mentioned in the manuscript,
Authors’ Response:
Abbreviations have been added.

Reviewer’s Comment:
In order to make it easier for the reader to follow the research hypotheses, it is worth considering to present in subsections 1.1.1,1.1.2,1.1.3 in figures the relationship/interrelationship between the particular components of the research model.

Authors’ Response:
figures have been added to each section.

Reviewer’s Comment:

It would be useful to highlight the relevant definition of terms in each of the subsections given above: i.e. patient satisfaction, patient loyalty, employee responsiveness
Authors’ Response:
definitions have been added.

Materials & Methods section:

Reviewer’s Comment:

Information should be added on the form in which the survey was conducted: online survey or questionnarie survey, and what was the sampling method

Please consider adding a table/statement with the research tools used, describing;

measured attribute: Service Quality; Patient Satisfaction; Patient Loyalty; Employee Responsiveness, Customer Citizenship Behavior

author of the tool, year of last version of the tool

names of dimensions/sub-dimensions, number of items of particular scale
Authors’ Response:
-sampling method is stratified sampling as mentioned in the methodology section and the collection online and face to face combined due to covid 19 restrictions (mentioned in the methodology section).
-table is added about the tools and scales with subdimensions used with authors.

Reviewer’s Comment:

Results section:

Age groups are not precisely defined, i.e., overlapping (e.g. respondents aged 25,35,45,55)

It is worth supplementing the results section with results from individual       scales concerning, for example, patient loyalty or patient satisfaction
Authors’ Response:
age groups were corrected. The mean scores were also added to the results section.

Discussion section:

Reviewer’s Comment:
Limitation of the study should be moved from the Conclusions section to the Discussion section.
Authors’ Response:
limitations and further research suggestions were moved to the discussion section.

Conclusions section:

Reviewer’s Comment:

It is important to emphasize the practical application of the results obtained.
Authors’ Response:
Practical implications have been added.

References:

Reviewer’s Comment:

The reference list needs formatting in accordance with the guidelines.
Authors’ Response:
references have been fixed.

Reviewer 3 Report

Thank you for the opportunity to review this study. My comments are as follows:

L84-L92: This paragraph does not belong to the introduction and has redundant information.

Section 1.1. Hypothesis Development: This could be replaced by a theoretical framework. Try to summarize the information into 3-4 paragraphs. Then, state the study hypotheses and the figure of the conceptual framework.

Section 2. Material & Methods: use subheadings such as design, Sample, Instruments, Data collection procedures, and Ethical Considerations. The information was difficult to follow.

How did you recruit the participant? Which method did you use to collect the data? Online?

3. Results: Use subheadings. Only the findings of the study aims should be presented. The findings related to psychometric testing should be reported in the methods as it is not one of the study aims.

This paper needs organization. There is much redundant information.

Author Response

Reviewer 3:

Thank you for the opportunity to review this study. My comments are as follows:

L84-L92: This paragraph does not belong to the introduction and has redundant information.

Reviewer’s Comment:
Section 1.1. Hypothesis Development: This could be replaced by a theoretical framework. Try to summarize the information into 3-4 paragraphs. Then, state the study hypotheses and the figure of the conceptual framework.
Authors’ Response:
The heading is updated and the information is summarized.

Reviewer’s Comment:
Section 2. Material & Methods: use subheadings such as design, Sample, Instruments, Data collection procedures, and Ethical Considerations. The information was difficult to follow.

How did you recruit the participant? Which method did you use to collect the data? Online?
Authors’ Response:
methodology section is updated as per the suggestions of the reviewer.

Reviewer’s Comment:
3. Results: Use subheadings. Only the findings of the study aims should be presented. The findings related to psychometric testing should be reported in the methods as it is not one of the study aims. This paper needs organization. There is much redundant information.
Authors’ Response:

The psychometric testing (reliability and validity statistics) were moved to the methodology section. Sub-headings have been added.

Round 2

Reviewer 3 Report

The psychometric evaluation should be presented briefly without tables. Again, the authors should follow the objectives of the study.

Author Response

Author's response for reviewer suggestion:

(psychometric evaluation is presented without the tables as suggested by the reviewer).
